# Intestinal microbiome and maternal mental health: preventing parental stress and enhancing resilience in mothers
Michiko Matsunaga [1,2,3] ✉, Mariko Takeuchi[1], Satoshi Watanabe[4], Aya K. Takeda[4], Takefumi Kikusui[5], Kazutaka Mogi[5], Miho Nagasawa[5], Keisuke Hagihara[1] & Masako Myowa [2] ✉

The number of mothers suffering from mental illness is increasing steadily, particularly under conditions of the coronavirus pandemic. The identification of factors that contribute to resilience in mothers is urgently needed to decrease the risks of poor physical and psychological health. We focused on the risk of parenting stress and psychological resilience in healthy mothers with no psychiatric and physical disorders and conducted two studies to examine the relationships between intestinal microbiota, physical condition, and psychological state. Our results showed that alpha diversity and beta diversity of the microbiome are related to high parenting stress risk. Psychological resilience and physical conditions were associated with relative abundances of the genera *Blautia*, *Clostridium*, and *Eggerthella*. This study helps further understand the gut–brain axis mechanisms and supports proposals for enhancing resilience in mothers.

In developed countries, the number of mothers with psychiatric disorders is likely to continue to increase. For example, the latest data in Japan showed that 28.7% of mothers have a high risk of depression[1]. Maternal psychiatric disorders have been shown to affect children's mental health and cognitive development[2,3]. Preventing maternal mental illnesses requires the identification of not only risk factors but also factors that enhance resilience, resulting in the protection of children's development.

Recently, the association between the intestinal microbiota and mental function has attracted attention. The intestinal microbiota affects the central nervous system through autonomic nerves, neuropeptides, hormones, and the immune system, and can be changed and improved by several factors, such as dietary and lifestyle habits and exercise[4,5]. Recent studies have shown that the intestinal microbiota is associated with psychological disorders (e.g., depression and anxiety)[6,7] and physical health[8,9]. For example, a recent systematic review found that patients with depression and anxiety disorders have increased inflammatory intestinal bacteria and fewer short-chain fatty acid (SCFA)-producing intestinal bacteria[10]. SCFAs are produced by gut microbiota-induced fermentation of dietary fiber. Among the SCFAs, butyrate-producing bacteria in particular have many beneficial effects on the intestinal environment, such as improving the immune function of the intestinal mucosa and inhibiting cancer cells[11].

Furthermore, physical health is a crucial consideration when evaluating the risk of mental illness. Mental illnesses, such as depression, anxiety, and stress are closely related to physical complaints, including fatigue and insomnia[12,13]. The Multidimensional Physical Scale (MDPS) is a multidimensional assessment of women's physical symptoms that has been used to demonstrate the deterioration of a woman's physical condition in the prediction of their mental illness risk[13]. Indeed, psychomotor changes are among the nine symptoms of major depressive disorder listed in the Diagnostic and Statistical Manual for Mental Disorders-Fifth Ed. (DSMV)[14]. Other physical correlates of the risk of psychiatric disorders include muscle mass (e.g., skeletal muscle mass index [SMI]) and muscle strength (e.g., handgrip)[15,16]. However, to date, there have been no comprehensive evaluations of the relationships between mental and physical health in young adults, especially mothers of small children, in whom child-rearing can induce considerable physical and mental stress. Nevertheless, thorough assessment and maintenance of physical health, including microbiota, can contribute considerably to mental health and well-being. Understanding the relationships between the intestinal microbiota and mental and physical health in mothers will allow for the development of methods for preventing and treating mental disorders, which use the intestinal microbiota (e.g., administering probiotics or improving dietary habits)[17].

[1]Department of Advanced Hybrid Medicine, Graduate School of Medicine, Osaka University, Osaka, Japan. [2]Graduate School of Education, Kyoto University, Kyoto, Japan. [3]Japan Society for the Promotion of Science, Tokyo, Japan. [4]Cykinso, Inc, Tokyo, Japan. [5]School of Veterinary Medicine, Azabu University, Kanagawa, Japan. ✉e-mail: paprika3c5@gmail.com; myowa.masako.4x@kyoto-u.ac.jp

https://doi.org/10.1038/s42003-024-05884-5　　　　　　　　　　　　　　　　　　　　　　　　　　　　　　　　　　**Article**

Among psychological risks, parenting stress is an important risk factor that increases the risk of psychiatric disorders (e.g., depression and anxiety disorders) and child abuse in mothers[18,19]. We focused on parenting stress in nonclinical mothers in this study because preventing and relieving excessive parenting stress is important before it progresses to psychiatric disorders. The relationship between the intestinal microbiota and maternal stress has been investigated mainly during pregnancy using the Perceived Stress Scale[20,21]. However, no studies have examined the relationship between parenting stress and the intestinal microbiota in nonclinical postpartum mothers. In mothers who are raising children, stress is related not only to their own factors but also to various other factors, including the characteristics of their children, the resources of the child-rearing environment, and their physical health. Therefore, evaluating parenting stress is effective using the Parenting Stress Index (PSI)[22,23], a global standard scale that can assess parenting stress from two aspects: enhanced stress caused by their children and the parents' own factors. Maltreatment of children aged 0–3 is high in Japan[24], and parental stress (as assessed by the PSI) can increase further in children aged 2–3[25] as they develop some independence of action and begin to explore and test boundaries. Therefore, investigating the relationship between parental stress and intestinal microbiota in healthy mothers raising young children can contribute to our understanding of resilience in this population, thereby helping to protect the mental health of both parents and their children.

To better understand the risk factors for parenting stress and determine the physical and psychological factors related to resilience in healthy mothers of young children, a multiple-index evaluation is necessary. This should include an analysis of the microbiota of the gut–brain axis. Particularly in the early postpartum period and among primiparous mothers, we should note that dynamic physiological and physical changes occur, such as neuroendocrine hormones (e.g., oxytocin)[26], autonomic nervous system, and body composition and muscle strength[27]. For example, oxytocin is one of the most important endocrine hormones and is closely associated with the expression of parenting behavior[28–30]. However, significant individual differences in levels of oxytocin have been reported and it can both stimulate the secretion of the stress hormone cortisol[31] and increase maternal emotional processing[26,32]. Therefore, the function of oxytocin continues to be debated, and further research is needed on the relationship between oxytocin and maternal stress. Individual differences in the responsiveness of the autonomic nervous system, particularly vagal activity, may be a further

important biomarker of, and contributor to, psychological resilience, although studies are limited[33]. The vagus nerve is an important neural pathway that connects the gut (including information regarding the status of its microbiota) to the brain. Any relationship between gut microbiota and mental health would be mediated by the vagus nerve[34]. Finally, it has been reported that mothers physically recover to the prenatal condition from 6–8 weeks to >6 months after childbirth[35,36]. However, the extent of this change has not been quantified, and its relationship to mothers' mental health is poorly understood. Physical conditions are also an important factor for psychological resilience in postpartum mothers.

In this study, as a first step, we focused on nonclinical Japanese mothers and conducted two studies. For Study 1, we analyzed 339 stool samples obtained from Japanese mothers caring for infants and toddlers aged 0–4 years and investigated the association between the intestinal microbiota and parenting stress. We hypothesized that mothers with a high risk of parenting stress have lower microbiota diversity and fewer SCFA-producing intestinal bacteria than healthy mothers. In this large-scale data analysis, we also compared the physical conditions between the parenting stress risk and healthy groups based on three measurements (i.e., sleep duration, sleep quality, and indices of the MDPS[13]). Here, we hypothesized that mothers at a high risk of parenting stress would have poorer physical conditions than healthy mothers. In Study 2, we specifically focused on 27 primiparous mothers in the early postpartum period (within 3–6 months). We hypothesized the following, but due to a lack of empirical data, we explored the relationship between their gut microbiota, physical and physiological function, and psychological resilience. We explored the relationships among their intestinal microbiota, physical and physiological functions, and psychological resilience. Regarding physical and physiological conditions, the following three factors were quantified, which were deemed important in terms of the gut–brain axis and the relationship between mental health and the intestinal microbiota, particularly in the early postpartum period: body composition and physical function, autonomic function, and oxytocin. We predicted that mothers with less robust physical functioning (i.e., those with poorer muscle strength, poorer motor functions, and poorer autonomic functions) would have higher parenting stress and/or lower psychological resilience. In particular, we expected vagal nerve activity to be an important factor related to both intestinal microbiota and psychological function. We predicted that oxytocin levels would be associated with parenting stress and/or psychological resilience but did not predict the direction (positive or negative) of this relationship. To the best of our knowledge, this is the first study to investigate the relationship of the intestinal microbiota to the risk of parenting stress and psychological resilience, as well as physical condition, in nonclinical mothers.

## Results
### Study 1: relationships between parenting stress risk and physical condition, dietary and lifestyle habits, and intestinal microbiota among mothers raising children aged 0–4 years
Participants exceeding any of the PSI cutoff values (i.e., a total score of 221 points, with the following subscale scores: 124 points for the parental aspect and 101 points for the child aspect) were classified in the parenting stress risk group. The results showed that 65 of the 339 mothers (19.17%) were at a high risk of parenting stress (Fig. 1).

Regarding physical condition, we found that mothers at a high risk of parenting stress had lower subjective sleep quality ratings and worse MDPS scores on all five dimensions than the healthy group ($p < 0.01$, $q < 0.01$) (Table 1). These MDPS results suggest that digestive system dysfunction, physical depression, and low female hormone function (impaired microcirculation) were related to the risk of parenting stress.

In analyzing the intestinal microbiota, we found that the alpha diversity was lower in the stress risk group than in the healthy group (Shannon α: $p = 0.026$ and $q = 0.019$ by the Mann–Whitney U-test). In terms of beta diversity, we found a significant difference in the UniFrac distance between the stress risk and healthy groups ($F = 1.94$, $p = 0.007$ [with unweighted UniFrac distance] and $F = 2.29$, $p = 0.003$ [with weighted UniFrac distance]

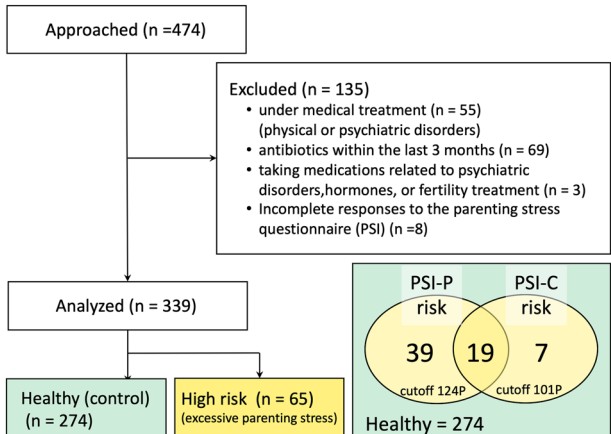

**Fig. 1 | Study 1 consort diagram and classification of psychological risk in mothers.** A consort flow diagram of Study 1. We finally analyzed data from 339 postpartum Japanese women. We evaluated the participants' mental health using the PSI with the standardized cutoff scores. Overall, 274 participants were classified into the healthy (control) group and 65 participants were classified into the risk group (i.e., parenting stress risk). The standardized cutoff for the PSI is 221 points, with subscale scores of 124 points for the parental aspect (PSI-P) and 101 points for the child aspect (PSI-C). Participants exceeding any of the PSI cutoff values were classified in the parenting stress risk group.

**Table 1 | Comparison of parenting stress and physical indices between the high parenting stress risk and healthy groups (Study 1)**

| | All participants (N = 339) | | Healthy Group (N = 274) | | Parenting Stress Risk Group (N = 65) | | Mann–Whitney U-test (Stress risk vs Healthy) | | | | |
|---|---|---|---|---|---|---|---|---|---|---|---|
| | Mean | SD | Mean | SD | Mean | SD | W | p-value | | q-value | |
| Parenting Stress (PSI)_total | 176.52 | 34.22 | 165.94 | 27.29 | 223.03 | 19.72 | 422.00 | 0.000 | ** | 0.000 | ** |
| Parenting Stress (PSI)_child | 76.17 | 16.48 | 72.54 | 14.67 | 92.12 | 14.52 | 2764.00 | 0.000 | ** | 0.000 | ** |
| Parenting Stress (PSI)_parent | 100.35 | 21.36 | 93.40 | 16.37 | 130.92 | 11.72 | 341.50 | 0.000 | ** | 0.000 | ** |
| Physical activity index (MDPS_PAI) | 3.07 | 1.54 | 2.80 | 1.49 | 4.22 | 1.22 | 4118.00 | 0.000 | ** | 0.000 | ** |
| Somatic disorders index (MDPS_SDI) | 0.97 | 1.21 | 0.82 | 1.09 | 1.60 | 1.50 | 6166.00 | 0.000 | ** | 0.000 | ** |
| Hormone activity index (MDPS_HAI) | 4.23 | 1.76 | 4.03 | 1.74 | 5.11 | 1.56 | 5898.00 | 0.000 | ** | 0.000 | ** |
| Microvascular disorders index (MDPS_MDI) | 3.63 | 2.00 | 3.43 | 1.97 | 4.49 | 1.92 | 6283.50 | 0.000 | ** | 0.000 | ** |
| Meteoropathy-related index (MDPS_MRI) | 2.70 | 1.66 | 2.57 | 1.60 | 3.26 | 1.76 | 6953.00 | 0.005 | ** | 0.007 | ** |
| sleeping time (hours) | 6.78 | 1.09 | 6.81 | 1.09 | 6.63 | 1.08 | 9773.50 | 0.189 | | 0.077 | † |
| sleeping quality | 1.91 | 0.60 | 1.95 | 0.61 | 1.75 | 0.50 | 10327.50 | 0.019 | * | 0.015 | * |

$**p$ or $q < 0.01$; $*p$ or $q < 0.05$; $†q < 0.10$.

by permutational multivariate analysis of variance [PERMANOVA]). Significant differences in the following bacteria were observed between the groups: *Odoribacter, Alistipes, Erysipelatoclostridium, Lachnospira, Monoglobus, Phascolarctobacterium, Veillonella, Sutterella,* and *Escherichia-Shigella* (Fig. 2 and Supplementary Table 1). These results were not affected by dietary habits because no significant group differences were observed between the groups (Supplementary Table 2). Moreover, these results were not affected by controlling for the mothers' age and education as covariates by post-hoc analysis of covariance (ANCOVA) test (Supplementary Table 3).

## Study 2: Body composition and physical function of primiparous mothers in the early postpartum period

To more objectively evaluate mothers' physical condition in the early postpartum period, particularly primiparous mothers, in Study 2, we quantitatively evaluated the following three factors, which are important in the association between maternal mental health and the intestinal microbiota from the perspective of the gut–brain axis: body composition and physical function, autonomic function, and oxytocin. First, we assessed the body composition and physical functioning (e.g., muscle strength) of the participants and referred to the medical diagnostic criteria or reference values for women of the same age from previous studies (Table 2). Figure 3a shows that the body mass index (BMI) was similar to the reference values for most participants. However, the SMI was lower than the medically diagnosed criterion for sarcopenia in nearly half of the participants ($n = 13$, 40.74%) (Fig. 3b). Figure 3c shows that most participants ($n = 23$, 85.19%) had lower hand grip strength than the reference values; hand grip strength reflects the total body muscle strength. Furthermore, Fig. 3d–f show that most participants showed lower values than the reference values for the two-step test for overall lower limb motor function ($n = 26$, 96.30%), normal gait speed ($n = 19$, 70.37%), and maximum gait speed ($n = 25$, 92.59%). These results indicate lower muscle mass and poorer motor function among Japanese mothers than among control women of the same age.

## Intestinal microbiota, autonomic nervous system functioning, and resilience in the early postpartum period (Study 2)

Next, we investigated the relationships between the intestinal microbiota, physical condition and physiological function (i.e., autonomic nervous system and oxytocin), and psychological resilience among primiparous mothers in the early postpartum period. First, we found a positive correlation between intestinal microbiota diversity (Shannon α) and vagal nervous activity ($r = 0.59$, $p = 0.003$, 95% confidence interval [CI]; 0.005–0.71) (Fig. 3g and Supplementary Figs. 1 and 2). Additionally, vagal nervous activity was positively correlated with psychological resilience (J-RS)

($r = 0.42$, $p = 0.047$, 95% CI 0.23–0.80) (Fig. 3h and Supplementary Figs. 1 and 2).

Next, shotgun metagenome and regression analyses of the mothers' microbiota showed that *Blautia SC05B48, Clostridium SY8519, Collinsella aerofaciens,* and *Eggerthella lenta* provided a significant partial explanation of the individual differences in psychological resilience and physical function (Fig. 4, Supplementary Figs. 2 and 3, and Supplementary Table 2). Specifically, *Blautia SC05B48* was positively associated with psychological resilience (J-RS) and two-step test score (Fig. 4A(1) and 4 A(2)). *Clostridium SY8519* was positively associated with psychological resilience (J-RS) and negatively associated with oxytocin (Fig. 4A(3) and 4 A(4)). *Collinsella aerofaciens* was positively associated with psychological resilience (J-RS) and negatively associated with maximum gait speed (Fig. 4A(5) and 4 A(6)). Furthermore, *E. lenta was* negatively associated with psychological resilience (RS25) and positively associated with parenting stress (child aspect subscale) and dominant hand grip strength (Fig. 4A(7), 4 A(8), and 4 A(9)). Finally, *Faecalibacterium prausnitzii* was positively associated with high sympathetic nerve activity among autonomic functions (Fig. 4A(10)). Figure 4B shows the significant results and Fig. 4C shows the null results, including statistics. Supplementary Fig. 3 shows the results of the identification of bacteria by shotgun metagenomic analysis.

## Discussion

A particularly noteworthy finding from these two studies was the association between three new factors and psychological stress and resilience in Japanese mothers: (1) the intestinal microbiota (e.g., Shannon α and β diversity, SCFA-producing bacteria), (2) autonomic nervous system function (e.g., vagal nervous activity) and neuroendocrine hormone (i.e., oxytocin), and (3) physical condition.

In Study 1, we examined the relationships between mental health risk (i.e., parenting stress risk) and the intestinal microbiota in a large sample of Japanese women caring for young children aged 0–4 years. We found that based on their PSI scores, ~19% of the mothers were at a high risk of parenting stress, although no participants were diagnosed with clinical psychiatric or physical illness. Mothers with high parenting stress had poorer sleep quality and physical condition assessed using the MDPS, including hormonal imbalance, and poor digestive function and blood circulation.

Indeed, the high parenting stress risk group had a lower Shannon α diversity of microbiota than the healthy group. Moreover, a significant difference in β diversity was observed between mothers with high parenting stress risk and healthy mothers, with three notable features in their prevalent microbiota: (i) abnormal levels of SCFA-producing bacteria (e.g., *Lachnospira, Veillonella, Alistipes,* and *Phascolarctobacterium*), (ii) fewer

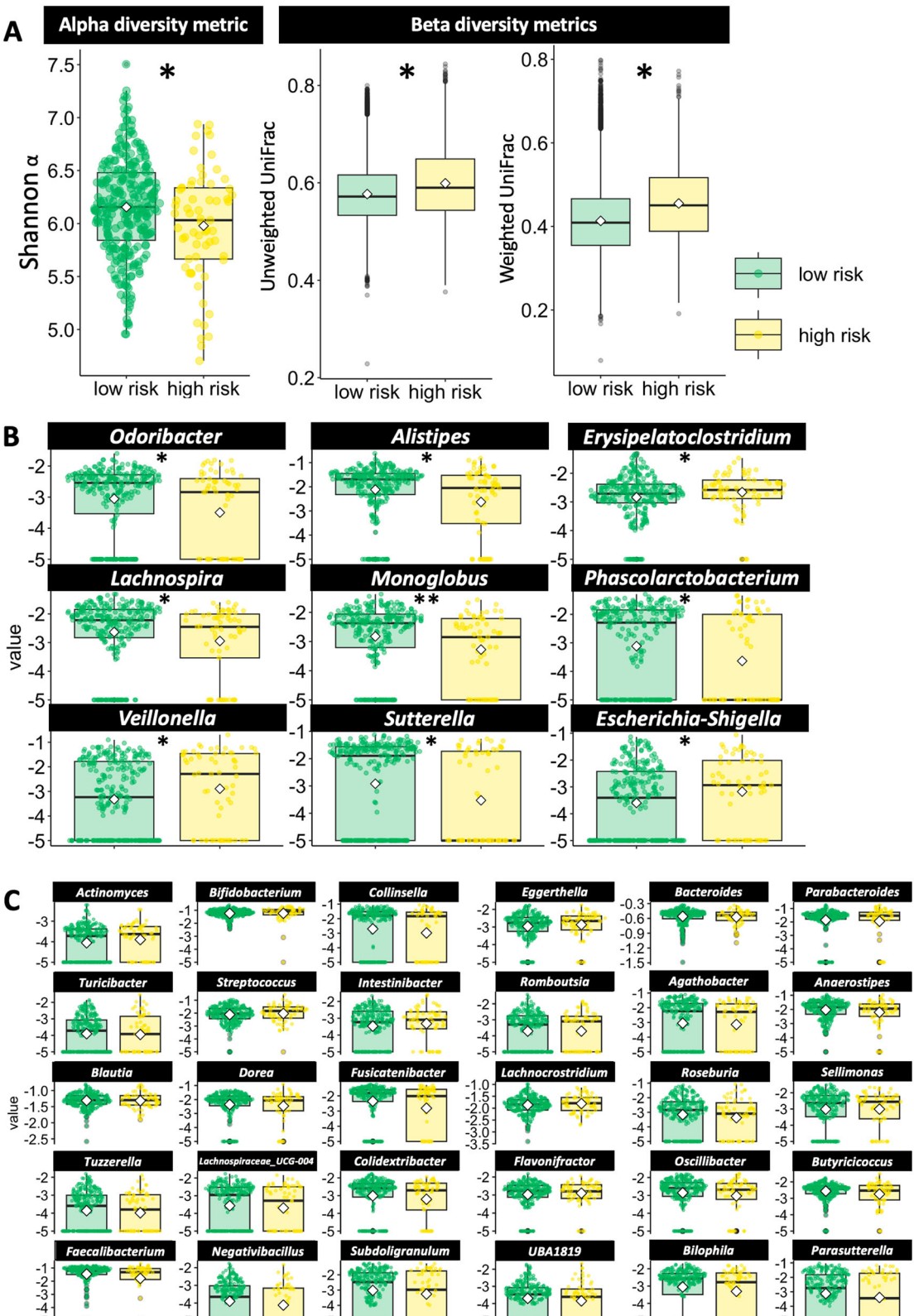

**Fig. 2 | Comparisons of the intestinal microbiota and parenting stress risk (healthy vs. high-risk groups) (Study 1). A** Results of the alpha and beta diversity metrics of intestinal microbiota. **B** Significant results of the relative abundance of genera. **C** Nonsignificant results of the relative abundance of genera. **\*\***p or q < 0.05.

intestinal bacteria related to immunity and antibiotic effects (e.g., *Odoribacter* and *Sutterella*), and (iii) more inflammatory intestinal microbiota (e.g., *Escherichia–Shigella*).

Regarding the first feature (i), abnormal levels of SCFA-producing bacteria have been reported by a previous study. *Lachnospira* is a butyrate-

producing bacterium that has many beneficial effects on the intestinal environment, such as improving the immune function of the intestinal mucosa and inhibiting cancer cells[11]. A recent systematic review also showed results consistent with our findings: fewer butyrate-producing bacteria in patients with depression[10]. We also found a significant difference in acetic

**Table 2 | Body composition and physical function among primiparous mothers and reference values (Study 2)**

|  | *Mean* | *SD* | **Min** | **Max** | **Reference value** |  |
|---|---|---|---|---|---|---|
| mother's age (years) | 33.63 | 4.18 | 27 | 41 | – |  |
| mother's education (years) | 16.19 | 1.39 | 12 | 19 | – |  |
| BMI (kg/m$^2$) | 20.54 | 2.67 | 15.4 | 26.80 | 21.6 kg/m$^2$ | Ministry of Health, Labour and Welfare, 2016[78] |
| FAT (%) | 26.49 | 5.68 | 14.9 | 37.30 | 24.70% | data provided by Inbody, Inc. |
| SMI (kg/m$^2$) | 5.89 | 0.59 | 5.00 | 7.20 | <5.7 kg/m$^2$ | diagnostic cliteria of salcopenia |
| ECW/TBW | 0.383 | 0.004 | 0.373 | 0.390 | 0.382 | Ohashi et al., 2018[79] |
| handgrip (kg) | 24.05 | 4.21 | 16.10 | 32.80 | 28.77 kg | Japan Sports Agency, 2017[80] |
| two-step test | 1.37 | 0.14 | 0.82 | 1.54 | 1.53 | Yamada et al., 2020[81] |
| normal gait speed (m/s) | 1.18 | 0.20 | 0.70 | 1.64 | 1.26 m/s | Obuchi et al., 2020[82] |
| maximum gait speed (m/s) | 1.92 | 0.38 | 1.08 | 3.00 | 2.5–2.8 m/s | Yokohama Sports Medicine Center, 2015 [83] |
| RS25 | 89.44 | 8.46 | 74 | 107 | – |  |
| J–RS | 93.93 | 13.61 | 69 | 115 | – |  |
| PSI_parent | 172.67 | 35.76 | 111 | 231 | – |  |
| PSI_infant | 98.89 | 22.32 | 53 | 144 | – |  |
| PSI_total | 73.78 | 19.03 | 46 | 108 | – |  |
| CESD | 7.81 | 4.59 | 0 | 18 | – |  |
| CVI | 4.25 | 0.25 | 3.77 | 4.62 | – |  |
| CSI | 2.95 | 0.82 | 1.4 | 4.33 | – |  |
| oxytocin (pg/ml) | 129.41 | 93.86 | 25.57 | 409.82 | – |  |

Reference values other than for the SMI are averages for women of the same age[78–83]. *BMI* Body Mass Index, *FAT* body fat percentage, *SMI* Skeletal Muscle mass Index, *ECW/TBW* extracellular water/total body water ratio, *RS25* the Resilience Scale 25; *J-RS* the Japan Resilience Scale, *CESD* the Center for Epidemiological Studies Depression Scale, *CVI* the Cardiac Vagal Index, *CSI* the Cardiac Sympathetic Index.

acid- and propionic acid-producing bacteria (e.g., *Veillonella*, *Alistipes*, and *Phascolarctobacterium*) between mothers at a high risk of parenting stress and healthy mothers. The hydrogen produced in fermenting dietary fiber is mainly consumed to produce acetic acid in the gut, particularly in Japanese individuals, whereas in Europe, the United States, and China, it is consumed for methane production[37]. Acetic acid suppresses inflammation in the intestine and promotes the repair of epithelial cell damage, and propionic acid contributes to energy and lipid metabolism in the colon[38,39]. However, some researchers reported that excessive acetic and propionic acids increased psychosomatic diseases and stress: higher concentrations of acetic and propionic acids produced by *Veillonella* were related to the severity of irritable bowel syndrome[40].

Regarding the second and third features (ii and iii), previous research has found associations between psychiatric disorders and immunity and inflammation-related intestinal bacteria. They have found that *Odoribacter* produced antibacterial "isoalloLCA," which suppresses the growth of intestinal bacteria that cause diarrhea and abdominal pain[41]. *Escherichia-Shigella* is a recognized major pathogen that causes diarrhea and is associated with the severity of anxiety disorders[42]. Studies on mice have shown that *Sutterella* was also associated with alpha-defensins, a type of immunity in the gut that fluctuates in response to psychological stress[43]. Considering these findings, our results suggest that Japanese mothers with high parenting stress had intestinal microbiota dysbiosis and increased intestinal inflammation.

Study 2 was designed to focus on primiparous mothers within 6 months of giving birth and examined factors contributing to their physical and psychological resilience. We found that vagal nerve activity was positively related to both the diversity of the intestinal microbiota (Shannon α) and psychological resilience (J-RS) in mothers. High vagal function at rest reflects the capacity for stress tolerance[44], and the vagal nerve is less active under stressful situations and is associated with stress hormones, such as cortisol[45]. To the best of our knowledge, only one study has examined the relationship between vagal activity and diversity of intestinal microbiota[34], and vagal activity and resilience, which is a study of trauma in US military

personnel[33]. We also showed that individual differences in vagal activity were a possible biomarker of psychological resilience; assessing individual differences in both gut microbiota and vagal activity may help to clarify the physiological factors that contribute to psychological resilience.

Among the prevalent intestinal microbiota, we found *Blautia SC05B48*, *Clostridium SY8519*, *Collinsella aerofaciens*, and *Eggerthella lenta* to be associated with individual differences in both psychological resilience and physical functions in primiparous women in the early postpartum period. *Blautia* and Clostridium are butyrate-producing bacteria, and *Blautia* in particular is more prevalent in Japanese individuals than those from Western and other Asian populations[46]. It has attracted attention for its potential positive effects on physical health (e.g., metabolic syndrome, lifestyle-related diseases, and healthy life expectancy)[47]. The results of this study provide new evidence of the relationships between individual differences in *Blautia* and psychological resilience in mothers. In this study, *E. lenta* was shown to be negatively correlated with psychological resilience (RS25) and positively correlated with parental stress and grip strength, consistent with previous findings of increased *E. lenta* in patients with major depressive disorder[48]. An important feature of *E. lenta* is its ability to metabolize isoflavones to produce equol, which has female hormone-like effects[49,50]. Equol has many benefits to women's health, including protection against breast cancer, osteoporosis, and arteriosclerosis. The Japanese diet features high soybean intake; however, only ~ 50% of the population has equol-producing microbes in the gut; in the United States and European countries, the proportions are ~20–30%[50]. Probiotics and dietary interventions focused on equol may be effective in reducing and even preventing parenting stress in Japanese mothers.

Additionally, *Clostridium* contributes to the suppression of intestinal inflammation and allergic responses by inducing the production of regulatory T cells that suppress immune system responses[51]. In this study, *Clostridium SY8519* was shown to be positively related to the J-RS score but negatively associated with oxytocin. This finding may be attributed to the function of oxytocin on the stress system. Under stressful conditions, the hypothalamic–pituitary–adrenal (HPA) axis is activated to secrete cortisol,

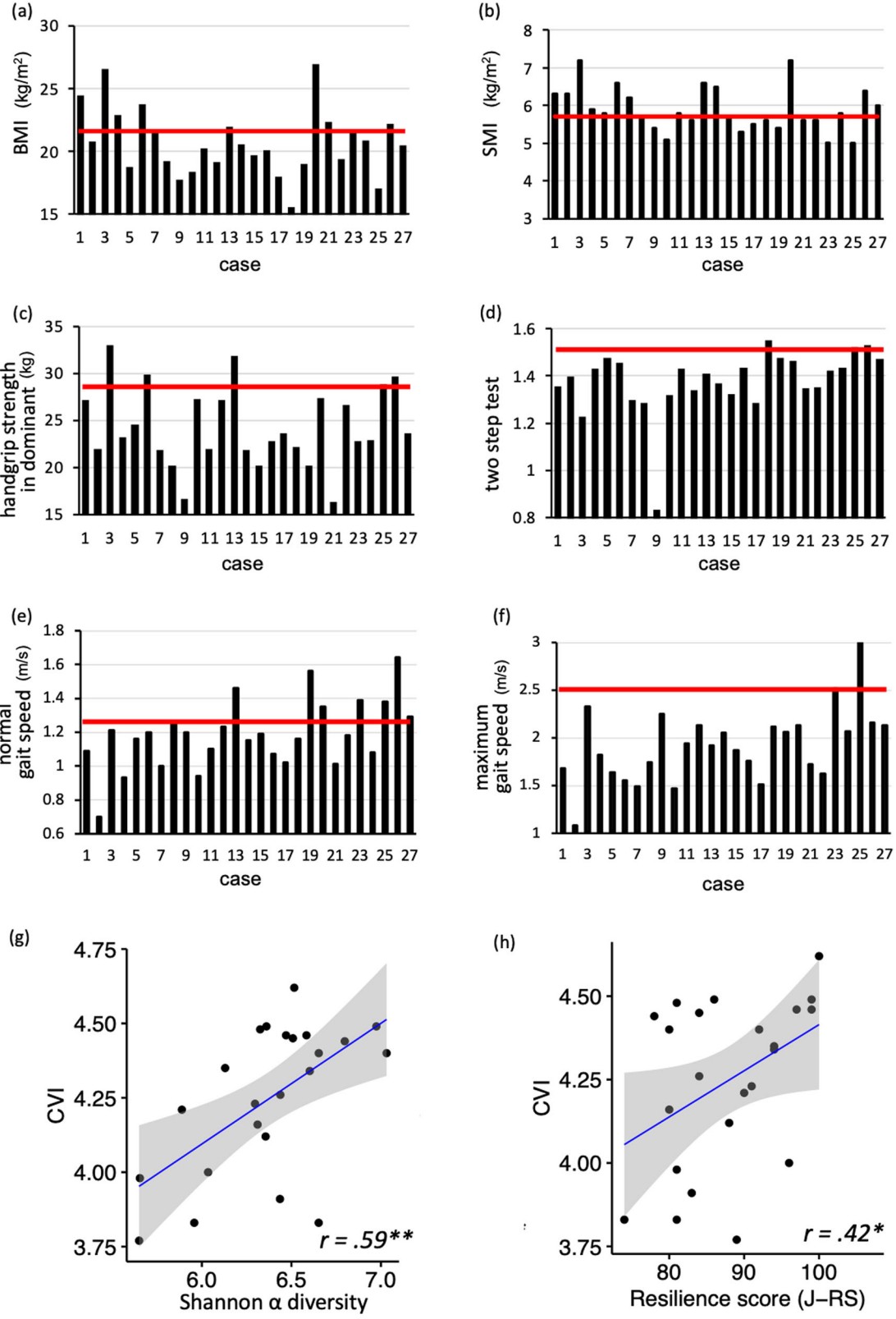

**Fig. 3 | Physical assessment of primiparous mothers in the early postpartum period (Study 2). a–f** show the physical condition of each participant. The horizontal axis (i.e., case) and black bars indicate representative values for each participant. Red lines indicate the reference values shown in Table 2. **g** Pearson's correlation between vagal nervous function (CVI) assessed by electrocardiogram and diversity of the intestinal microbiota (Shannon $\alpha$). **h** Pearson's correlation between CVI and resilience (J-RS) scores, as assessed by questionnaires. Gray bands in (**g**) and (**h**) show 95% confidence intervals, and each dot represents a single participant. **$p < 0.01$, *$p < 0.05$.

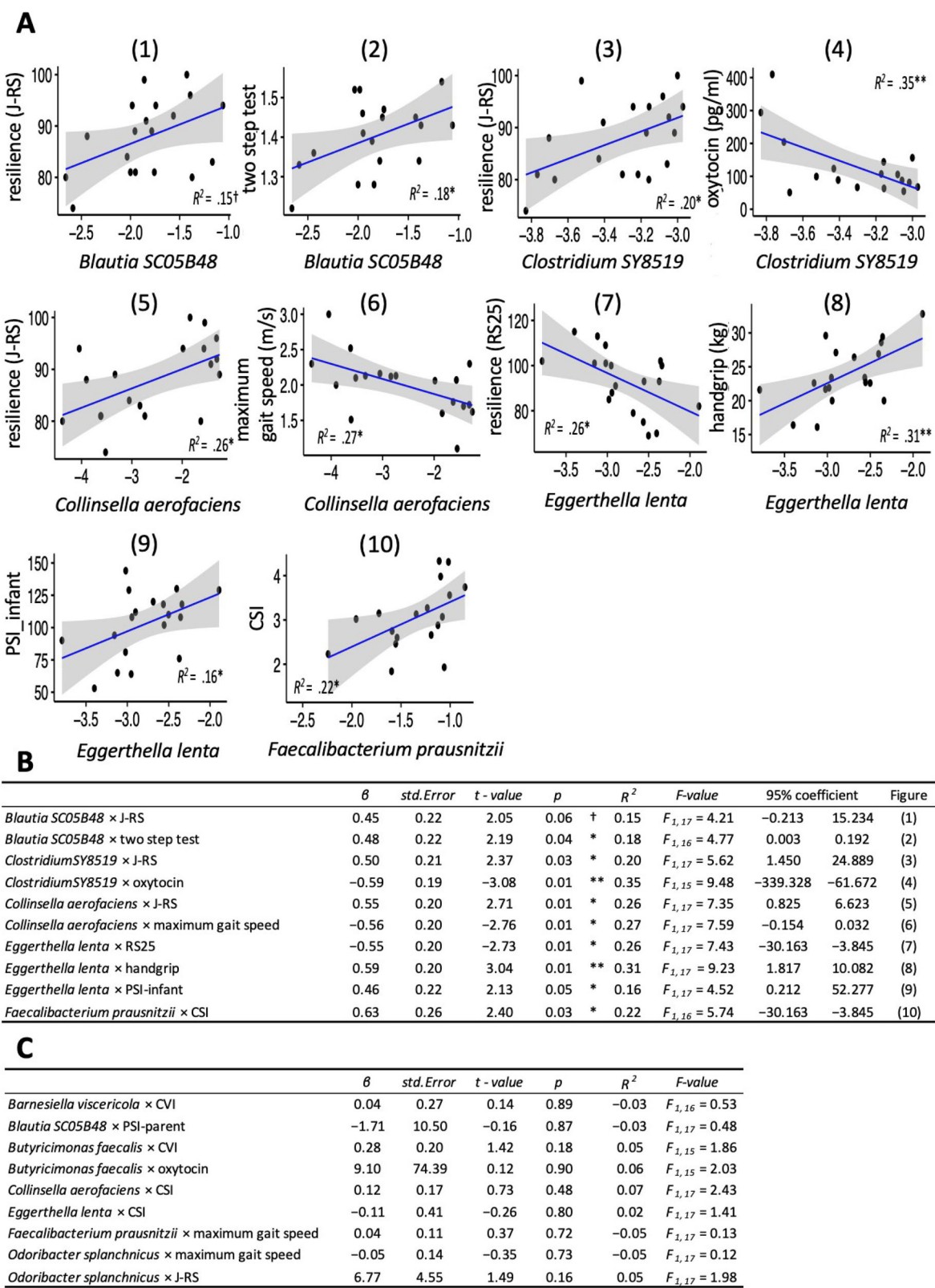

**Fig. 4 | Relationships between individual differences in the intestinal microbiota and physical and psychological resilience.** Scatter plots and regression lines (**A**) and statistical results (**B**) of intestinal microbiota with 95% CIs show significant relationships between the intestinal microbiota (i.e., horizontal axis) and physical and psychological indices (i.e., vertical axis). Each dot represents a single participant. In the regression analysis of *Blautia SC05B48* and the two-step test, we excluded one participant from the statistical analysis because her two-step test score was an outlier (mean minus 3 *SD* or less). Results of nonsignificant relationships (**C**). J-RS the Japan Resilience Scale, RS25 the Resilience Scale 25, PSI infant child aspect of the PSI (subscale of PSI), CSI the Cardiac Sympathetic Index, CVI the Cardiac Vagal Index, $\beta$ standardized partial regression coefficient, **$p < 0.01$, *$p < 0.05$, †$p = 0.06$ (effect size > medium [$R^2 > 0.13$]).

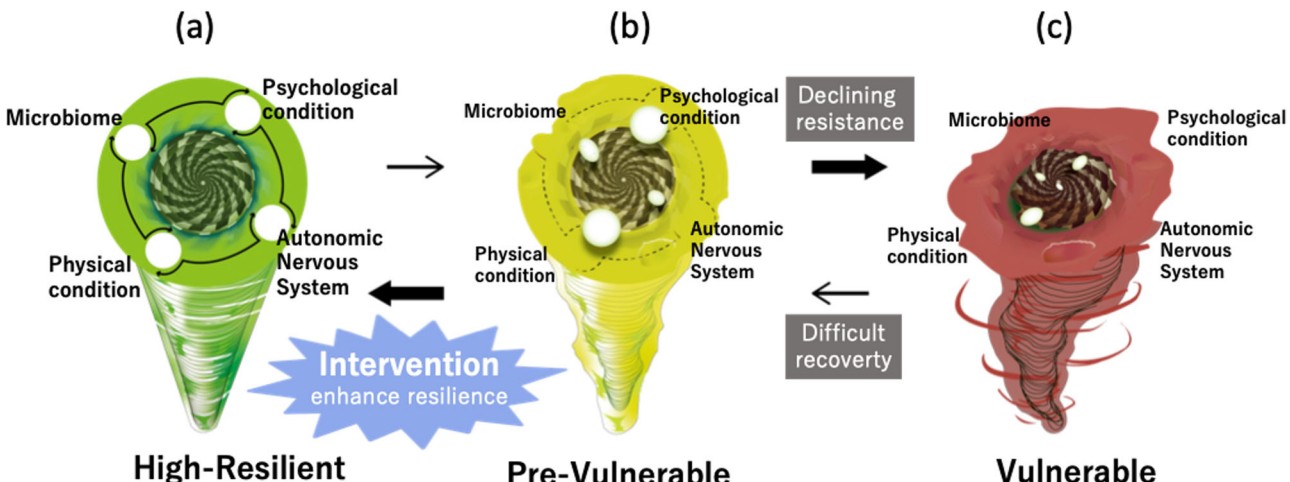

**Fig. 5 | Putative conceptual model of physical and psychological resilience. a** High resilience state in which all physical and psychological factors are healthy. **b** Pre-vulnerable state, characterized by the onset of decline in physical and psychological functions, before falling into a vulnerable state. **c** Vulnerable state, with deterioration of physical, and psychological condition.

and simultaneously, the oxytocin system in the hypothalamic para-ventricular nucleus is activated[52,53]. Furthermore, oxytocin secretion was shown to suppress cortisol secretion[31]. Therefore, a state of high oxytocin means that there is a high physiological need for immune or stress responses on the HPA axis to restore homeostasis. Our results showed that mothers with high *Clostridium SY8519* had low oxytocin levels and high psychological resilience. One interpretation of these results is that mothers with high *Clostridium SY8519* and low oxytocin levels may have stable homeostasis, which allows for higher psychological resilience. Higher oxytocin levels in new mothers have positive effects on the mother and child[26,54,55]. Future studies should aim to examine and clarify the relationships between intraindividual variations in oxytocin levels, stress responses, and intestinal microbiota.

As limitations of the study, further research is required in the two following aspects from the results of our study. First, regarding physical functioning in Study 2, we found that ~ 41% of the primiparous mothers had SMI of <5.7 kg/m2, which met one of the criteria for sarcopenia. Upper extremity muscle strength and lower extremity motor function were also lower in most mothers than the reference values for women of the same age. The SMI peaks at ~20–30 years of age and then declines by 1–2% per year[56], and age-related decline in the SMI can ultimately lead to frailty and sarcopenia. To make conclusions about the relationship between physical condition and mental function, assessing the degree of decline and recovery of muscle mass and physical functions from prepregnancy to postpartum is important in future studies.

The second point that needs further study is that no group differences in dietary habits were observed between the high-risk and healthy groups in Study 1. However, the results of the microbiota (i.e., *Erysipelatoclostridium* in Study 1 and *Blautia SC05B48* and *Clostridium SY8519* in Study 2) suggest that the Japanese diet can prevent or enhance resilience in mothers. For example, *Erysipelatoclostridium* has been shown to improve lipid metabolism when administered with green tea leaf powder in a mouse study[57]. *Clostridium SY8519* is involved in the process of metabolizing isoflavones[58]. Isoflavones are among the flavonoids, and the Japanese diet traditionally includes various foods rich in flavonoids (e.g., green tea, soybeans, sesame, and yuzu (a small citrus fruit)). Regarding *Blautia*, studies on mice have revealed that glucosylceramide, a component of koji, increased *Blautia*[37]; koji is an ingredient used in most Japanese traditional fermented foods (e.g., soy sauce, miso, sake, and vinegar). Several researchers have reported that dietary interventions with glucosylceramide derived from soybean and rice bran reduced the incidence of colon, head, and neck cancers[59,60]. With further research, the Japanese diet may be an approach for improving the intestinal microbiota and buffering mothers from excessive parenting stress.

Whether the results of this study are unique to data obtained in Japan or common worldwide is an important question that should be examined in future studies, both by accumulating further research on the Japanese and by conducting international comparisons.

This study had several limitations. We collected the data for Study 2 before the wider effects of coronavirus-2019 had hit. Although we had planned to include a larger sample size, we had to discontinue the study and perform preliminary analyses based on data from 27 cases only. In Study 2, a regression analysis was performed to examine the relationship between intestinal microbiota and physical/physiological and psychological states. To prevent unnecessary statistical tests, we performed several correlation analyses to preprocess the regression analysis data. Owing to the small sample size, we did not perform $p$-value correction or mediation analysis considering the possibility of type 2 error and failure to perform appropriate $p$-value correction. Although we acknowledge the possibility of a type 1 error, all results had moderate to large effect sizes, and our single regression analysis confirmed significant associations. We also reported all non-significant results and provided scatter plots and details of 95% confidence intervals. Another issue that arose due to the small sample size was the inability to include covariates in Study 2. In addition, Study 2 participants had lower average parenting stress and depression scores than Study 1 participants. As both Study 1 and Study 2 were cross-sectional, we could not evaluate intrapersonal changes before and after childbirth. Therefore, longitudinal studies with a larger sample sizes, including samples from clinical groups with high stress levels, are required to clarify the neuro-physiological and psychological relationships. In future research with larger samples, it would be helpful to examine covariates, such as the body composition, dietary habits, and exercise habits of the mothers, to better understand the physical and mental effects of the intestinal microbiota.

In summary, we preliminarily investigated relationships among the following four variables; intestinal microbiota, autonomic nervous system function, physical condition (e.g., body composition, physical function, and physical condition), and psychological state (Fig. 5a–c). The intestinal microbiota of mothers with high parenting stress was in a vulnerable state (Fig. 5c): decreased diversity and butyrate-producing bacteria (e.g., *Lachnospira*), suggesting the dysbiosis of the intestinal environment. Particularly for early postpartum phase and primiparous mothers, the intestinal microbiota, and autonomic nervous system function are related to physical and psychological resilience. A weakened autonomic nervous system and compromised physical functions may impede resilience and recovery and lead to prolonged physical and mental illness. Screening of mothers at high risk of stress is important (Fig. 5b), allowing intervention to enhance their resilience before they become vulnerable (Fig. 5c). Because the composition

**Table 3 | Characteristics of Study 1 participants**

|  | Mean | SD | range |
|---|---|---|---|
| Mother's age (years) | 34.66 | 4.81 | 21–47 |
| Education (years) | 14.61 | 1.98 | 9–22 |
| **Household income** | **N** | **%** |  |
| <3 million yen (about 26,000 USD) | 36 | 10.62 |  |
| 3–5 million yen (about 26,000–43,360 USD) | 86 | 25.37 |  |
| 5–7 million yen (about 43,360–60,700 USD) | 92 | 27.14 |  |
| >7 million yen (about 60,700 USD) | 101 | 29.79 |  |
| no answer | 24 | 7.08 |  |
| **Number of children** | **N** | **%** |  |
| one | 71 | 20.94 |  |
| two or more | 231 | 68.14 |  |
| pregnant child (2nd or 3rd or 5th child) | 24 | 7.08 |  |
| no answer | 13 | 3.83 |  |
| **Age of youngest child (Years since last childbirth)** | **N** | **%** |  |
| less than 1 year old | 41 | 12.09 |  |
| 1–2 years old | 156 | 46.02 |  |
| more than 3 years old | 126 | 37.17 |  |
| no answer | 16 | 4.72 |  |

of the intestinal microbiota differs among ethnic populations[37,46], the association between the intestinal microbiota and resilience has some ethnic-specific aspects. This suggests that effective support and healthcare should be developed in a tailor-made manner considering ethnic traits and individual differences. In conclusion, our study helps further understand the microbiota of the gut–brain axis[4] underlying individual differences in mental health and proposes personalized interventions that can enhance resilience.

## Methods

### Participants and procedures (Study 1 and 2)

Study 1 is part of a research project in Japan entitled, "*The Principles of Human Social Brain-Mind Development.*" The project aims to create a database of intestinal microbiota from >1000 mother-child pairs in Japan and to investigate the relationships between those microbiota and emotional and cognitive development. Participants were recruited through nursery schools throughout Japan. Study 1 reports on intestinal microbiota data collected between January and February 2021. Figure 1a presents a consort diagram of the data collected from 474 postpartum women raising children aged 0–4 years. Of these 474 participants, we excluded 135 for the following reasons: (i) receiving medical treatment for a physical or psychological disease (e.g., diabetes, irritable bowel syndrome, depression) at the time of data collection; (ii) having taken antibiotics within the last 3 months; (iii) taking hormones, medications for psychiatric disorders, or fertility treatments (i.e., escitalopram, sertraline, chlormadinone acetate, lithium carbonate, or levothyroxine sodium hydrate); and (iv) incomplete responses to the parental stress questionnaire. Finally, we analyzed data from 339 participants (mean age, 34.66 years; SD, 4.81; range, 21–47 years). Table 3 shows the characteristics of our Study 1 participants. All participants had completed at least 9 years of education, and their annual household income ranged from <3000,000 yen (~ 26,000 USD) to >15,000,000 yen (~ 130,000 USD). Around 52% of respondents reported an annual income of >7,000,000 yen (~ 60,700 USD). The average household income of Japan's child-rearing generation in 2019 was 7,459,000 yen (~ 64,700 USD)[61]. Hence, >60% of the participants in Study 1 had a below-average annual household income.

We obtained written informed consent from each participant before collecting a stool sample and providing them with questionnaires to complete at home. We analyzed three questionnaires in this study; the PSI[22,23], MDPS (physical condition)[13], and Mykinso Pro (dietary and lifestyle habits). Furthermore, we collected information on socioeconomic status. See questionnaires (Studies 1 and 2) in this manuscript for more details. This study was approved by the Medical Ethics Committee of Kyoto University (no. R2624) and registered in the UMIN system (UMIN000042508). All ethical regulations relevant to human research participants were followed.

Study 2 is part of the Japanese research project entitled, "*Investigation of the Interrelationship Among Mental, Physical, and Intestinal Microbiota in Postpartum Mothers and Their Infants.*" For this study, we collected and analyzed data from 27 first-time mothers within 3–6 months after childbirth (mean age, 33.63 ± 4.18; range, 27–41 years). The average postpartum duration was 152.52 days (SD, 19.1 days; range, 115–191 days). None of the participants had a psychiatric diagnosis. All 27 participants continued to breastfeed their infants for the duration of the study period: 15 (55.56%) breastfed exclusively, and 12 (44.44%) supplemented breastmilk with formula. Of the 27 women, 22 (81.48%) gave birth vaginally, and five (18.52%) via cesarean section. Eleven women (40.74%) used antimicrobials during delivery. We collected the data between September 2019 and March 2020. All participants visited the Baby Laboratory at Kyoto University for the study and provided written informed consent. This study was approved by the Medical Ethics Committees of Osaka University (no. 18409 and no. 20074) and Kyoto University (no. 27-P-1) and registered in the UMIN system (UMIN000045125). All ethical regulations relevant to human research participants were followed.

The participants visited the lab twice. At the first visit, we collected the physical data and administered the questionnaires: the Japanese version of the Resilience Scale (RS25 and J-RS, psychological resilience) and the Center for Epidemiological Studies Depression Scale (CESD). The physical measurements used to assess physical composition and function were as follows: (1) BMI, body fat percentage, SMI, and extracellular water/total body water ratio (ECW/TBW) (InBody 770 multifrequency bioelectrical impedance device, InBody Japan Inc., Tokyo); (2) grip strength of the dominant hand, which is considered to reflect the muscular strength of the entire body; (3) the two-step test for the risk of locomotive syndrome (a lower score indicates weak gait function, including muscle strength, balance, and flexibility of the lower limbs); (4) normal and maximum gait speed for 5 m (a lower gait speed score also indicates declining motor function in the lower limbs).

At the second visit, we collected saliva samples to measure oxytocin, collected 3 min of resting electrocardiogram data to assess the autonomic nervous system (i.e., cardiac sympathetic and cardiac vagal index; based on Toichi et al. [1997])[62], administered the PSI, and collected information on age, years of education, and breastfeeding schedule. Although we instructed the participants to visit twice within the same week if they could, the mean interval between the two visits was 7.74 ± 8.19 days. To evaluate the intestinal microbiota, the participants were instructed to collect stool samples at home within 2 or 3 days after the second visit. See questionnaires (Studies 1 and 2), and physical and physiological measurements in Study 2 in this manuscript for more details.

### Questionnaires (Studies 1 and 2)

For Study 1, questionnaires were administered to the Japanese mothers that measured parenting stress, physical health, and dietary habits. We measured parenting stress using the Japanese version of the Parenting Stress Index (PSI), a questionnaire developed in the United States[2] and subsequently validated and standardized in Japanese[23]. The PSI consists of 78 items and requires respondents to rate each item on a five-point scale. It includes two subscales: stress related to the characteristics of the child (i.e., the child aspect) and stress related to the parents themselves (i.e., the parent aspect). A total score denoting high parenting stress is ≥ 221, whereas child and parent scores denoting high parental stress are ≥ 101 and ≥ 124, respectively.

To assess the physical health of our participants, we used the Multi-dimensional Physical Scale (MDPS)[13], which is based on Oriental medicine criteria for physical pathology in postpartum women. The MDPS consists of 17 items covering five subscales, with each item rated on a three-point scale

(Supplementary Table 2). The subscales are a physical activity index (higher scores indicate greater physical inactivity [e.g., due to tiredness and lethargy]); (ii) a somatic disorders index (higher scores indicate more symptoms of physical depression [e.g., anorexia, bloating]); (iii) a hormone activity index (higher scores indicate reduced (female) hormone function [e.g., coldness, dizziness, dry skin]), (iv) a microvascular disorders index (higher scores indicate impaired microcirculation due to (female) hormone function [e.g., skin pigmentation and rough skin]); and (v) a meteoropathy-related index (higher scores indicate impaired water metabolism and meteorological diseases [e.g., swelling, headaches during bad weather]).

The participants' dietary and lifestyle habits were assessed using the Mykinso Pro Questionnaire, which includes questions on lifestyle (smoking, drinking, sleep, and defecation), dietary intake, and physical diseases (history of medical treatments and medication). The questionnaire was developed by Cykinso Co., a provider of intestinal microbiota screening services in Japan. To assess sleep time and sleep quality, participants were asked to report their average number of hours of sleep per night and to rate their sleep quality on a three-point scale (3 = good, 2 = fine, 1 = bad). To assess dietary intake, the questionnaire considers 18 food types. These are staple foods (i.e., rice, bread, and noodles), unrefined grain, root vegetables, green and yellow vegetables, light-colored vegetables, fruit, meat, fish and shellfish, eggs, milk and cheese, yogurts and lactic acid bacteria beverages, soybeans and soybean-derived foods, natto (fermented soybeans), seaweed, pickles, mushrooms, snacks, and sugary drinks. The questionnaire asks participants how frequently they consumed each food type over the preceding week of stool collection. Responses are given on a five-point scale, as follows: 0 = did not eat; 1 = 1–3 times/week; 2 = 4–6 times/week; 3 = once/day; 4 = twice or more/day. The questionnaire also collects sociodemographic information (age, years of education, income level, etc.).

In Study 2, we administered four questionnaires to assess parenting stress, depression, psychological resilience, and sociodemographic characteristics. As in Study 1, we measured parenting stress with the PSI. We measured depression using the Center for Epidemiological Studies Depression (CESD) scale, which was developed by the National Institute of Mental Health[63] and standardized for Japanese use[64]. The CESD scale consists of 20 items, each rated on a four-point scale. The cutoff score for depression is 16 or higher. To assess psychological resilience, we used the Resilience Scale (RS25)[65,66] and a Japanese version of the scale (J-RS). The RS25 is a validated, reliable, and popular quantitative resilience assessment scale that is widely used in Europe and the United States. It consists of 25 items, each rated on a seven-point scale. Researchers have attempted to validate a Japanese version of the RS25, but it did not have sufficient reliability or validity[66]. The researchers concluded that the scale might not be suitable for Japanese people because of the Japanese tendency to refrain from positive, proactive expressions. Resilience scales developed in Japan, such as the Mental Resilience Scale and the Two-Dimensional Resilience Factors Scale, have only been validated with students and young adults. For this study, we used both the RS25 and the J-RS. The J-RS has been validated in adults and ensured the inclusion of items related to characteristics valued in Japan, such as empathy, emotional control, and social connections.

## Physical and physiological measurements in Study 2

The body composition and physical function of Japanese new mothers were assessed using four tests. These were body composition, grip strength, the two-step test, and gait speed. Body composition was assessed using the InBody 770 multifrequency bioelectrical impedance (BIA) device (InBody Japan Inc., Tokyo, Japan). For this test, the participant stood on a balance scale, grasped the handle of the machine, and remained still for 1 min while maintaining a relaxed upright posture. Using data collected by the BIA device we recorded each participant's body weight, BMI, body fat percentage, SMI, and extracellular water/total body water ratio (ECW/TBW). We calculated BMI, body fat percentage, and SMI using the following formulas:

- Body mass index (BMI) (kg/m$^2$) = weight (kg)/square of height (m$^2$)
- Body fat percentage = fat mass (kg)/body weight (kg) × 100

- Skeletal muscle mass index (SMI) (kg/m$^2$) = skeletal muscle mass (kg)/square of height (m$^2$)

Grip strength in the dominant hand was measured using a digital grip strength meter (TKK-5401, Matsuyoshi Medical Instrument Co., Ltd., Tokyo, Japan). The participants grasped a steering wheel device with their arms stretched out to the side and lowered. We measured both hands twice, alternating between them, and used the highest value for the dominant hand in our analysis.

The two-step test measures the distance between two steps taken with as large a stride as possible[66,67]. Scores are calculated using the following formula:

Maximum stride length (cm)/height of individual (cm)

For each participant, we took two measurements and used the average.

We measured the maximum and normal gait speed[68] of each participant. This was achieved by using a stopwatch to measure the time they took to walk a distance of 5 m on an 11 m walking course (the additional 6 m allowed for acceleration and deceleration on a flat floor). Maximum gait speed was the speed measured when the participant walked as quickly as she could. Normal gait speed was measured with the participant walking at her normal pace. We took each measurement twice. For normal gait speed, we used the average of the two measurements. For maximum gait speed, we used the faster of the two measurements.

To measure individual differences in autonomic nervous system function, we performed electrocardiograms (ECG) on participants using the Polymate Mini AP108 (Miyuki Giken Co., Ltd., Tokyo, Japan). The ECG were recorded at 1000 Hz, with electrodes attached at the V2 point of the three-point induction. The R-R interval of the infants was calculated using the last minute of ECG data from the 3 min resting period. During these resting measurements, the mother sat on the floor and the infant lay on the floor. The mother was asked to relax and spend time with her infant as she usually would at home. Respiration was not measured. We analyzed the cardiac vagal and sympathetic indices using a Lorenz plot analysis method[62]. Lorenz plots are more robust than frequency analysis for analyzing ECG data over a short period or in the presence of noise due to body movement. They also have the advantage of allowing separate calculation of sympathetic and vagal activities and are independent of respiration and posture.

Prior to saliva collection for oxytocin hormone analysis, participants rinsed their mouths with water and, 10 min later, chewed on an oral cotton swab (Salimetrics, State College, PA, USA) placed sublingually for 3 min to collect saliva. We immediately stored the collected saliva at −80 °C. The process was then repeated to obtain a backup sample. We used the first samples to calculate the hormone levels of each participant. The second sample was used if the first contained insufficient saliva or had outlier hormone levels. Oxytocin concentrations were measured using a commercial kit (ADI-900-153A-0001; Enzo Co. Ltd., Tokyo, Japan) following the manufacturer's protocol. In brief, we dried 240 μL of saliva at room temperature for 3 h using a Speedvac evaporator. It was then reconstituted in 240 μL of assay buffer, from which 100 μL were used for the assay.

## Fecal sample collection and DNA extraction of intestinal microbiome analysis (Studies 1 and 2)

Fecal samples were collected using Mykinso fecal collection kits containing guanidine thiocyanate solution (Cykinso, Tokyo, Japan), transported at ambient temperature, and stored at 4 °C. DNA from the fecal samples was extracted using an automated DNA extraction machine (GENE PREP STAR PI-480, Kurabo Industries Ltd, Osaka, Japan), according to the manufacturer's protocol.

## 16 S rRNA gene sequencing (Studies 1 and 2)

The detailed sequencing methods are described elsewhere[69]. Briefly, amplicons of the V1V2 region were prepared using the forward primer (16S_27Fmod: TCG TCG GCA GCG TCA GAT GTG TAT AAG AGA CAG AGR GTT TGA TYM TGG CTC AG) and the reverse primer

(16S_338R: GTC TCG TGG GCT CGG AGA TGT GTA TAA GAG ACA GTG CTG CCT CCC GTA GGA GT). The libraries were sequenced in a 250-bp paired-end run using the MiSeq Reagent Kit v2 (Illumina; 500 cycles).

## Metagenomic shotgun sequencing (Study 2)
DNA for each sample was sheared to ~ 300 bp using Covaris ME220 (Covaris, Woburn, MA, USA). The shotgun sequencing library was constructed using the KAPA Hyper Prep Kit (Roche). Libraries were sequenced on DNBSEQ-G400 (BGI, Shenzhen, China) at a 150-bp paired-end.

## Bioinformatics analysis (1) 16 S rRNA analysis (Study 1)
The data processing and assignment using the QIIME2 pipeline (version 2020.8)[70] was performed as described elsewhere[71]. Briefly, paired-end reads were processed using DADA2[72], and taxonomy classifications for amplicon sequence variants were performed using a naïve Bayes classifier in QIIME2. The classifier was trained with arts-SILVA, a simplifier for the SILVA[73] 138 16 S rRNA taxonomy dataset, designed to enhance comprehensibility by refining taxonomic assignments and curating mis-entries in the SILVA database. Specifically, arts-SILVA removes labels with minimal information, corrects duplicate entries, and excludes irrelevant taxa such as "human metagenome" through manual inspection. It assigns a consensus taxonomy to each cluster only when there is 100% agreement between the assigned taxa, and eradicates the label "ambiguous taxa" to ensure a clear and concise taxonomy for analysis.

## Bioinformatics analysis (2) 16 S rRNA analysis (Study 2)
The data processing and assignment based on the QIIME2 pipeline (version 2019.4)[70] was performed in the following steps: (1) DADA2[72] for joining paired-end reads, filtering, and denoising and (2) assigning taxonomic information to each ASV using a naïve Bayes classifier in the QIIME2 classifier with the 16 S rRNA gene V1V2 region data from Greengenes[74,75] to determine the identity of bacteria and their composition.

## Bioinformatics analysis (3) Metagenomic shotgun analysis (Study 2)
Each read pair was mapped against a custom database, including human genome (hg19), bacteria and fungi in RefSeq, and viruses in nucleotide collection (NCBI-nt) by Kraken2[76].

## Bioinformatics analysis (4) Statistical analysis (Studies 1 and 2)
We performed all data manipulation, analyses, and graphics using R and RStudio (versions 3.5.1 and 1.1.456, respectively) and used the R package qiime2R and microbiome R for all analyses. We used the R package tidyMicro (version 1.48) and ggplot2 for visualizations. Regarding the evaluation of the diversity of the intestinal microbiota, we measured the alpha diversity at the ASV level using the Shannon index and the beat diversity using the unweighted and weighted UniFrac distances. Unweighted UniFrac distance means beta diversity incorporating taxa presence/absence. Weighted UniFrac distance means beta diversity incorporating the relative abundance of taxa. Additionally, we analyzed the composition of the prevalent microbiota as the set of genus-level bacterial groups shared by at least 50% of the participants, with a relative abundance of at least 0.01%, based on previous studies[77]. We calculated the relative abundances as the number of sequencing reads of each taxon in a sample standardized by the total number of sequences generated for that sample. In the analyses, we only included taxa present in at least one sample, and we aggregated sequence counts for taxa that did not meet these requirements into the "other" category. We applied these filters at the phylum, family, and genus levels and left sequence counts that we could not classify to the taxonomic level of interest as unclassified counts of the lowest level possible. See Supplementary Tables 4 and 5 for more details on the prevalent microbiomes detected in Studies 1 and 2.

## Statistics and reproducibility
We performed statistical analyses and visualizations for both studies using R and RStudio (versions 4.0.5 and 1.4.1106, respectively). We performed $q$-value estimation using R and RStudio (versions 4.2.1 and 1.4.1106, respectively) using the R package Bioconductor (version 3.15). For all statistical analyses, two-tailed tests were used.

In Study 1, We used the Mann–Whitney $U$-test to compare the PSI high-risk and healthy groups in terms of physical state score, dietary and lifestyle habits, and intestinal microbiota (i.e., alpha diversity, and the prevalent microbiota). We calculated the false discovery rate-adjusted $q$ for all indices. Regarding the group comparison in terms of the beta diversity of the intestinal microbiota, we conducted PERMANOVA.

In Study 2, we aimed to investigate the relationships among the intestinal microbiota, physical and physiological function, and psychological resilience in early mothers. First, we performed an exploratory correlation analysis and focused on the variables for which $p < 0.05$ or those with a moderate effect size ($r \geq 0.30$). Next, to clarify the extent to which individual differences in the intestinal microbiota could explain individual differences in psychosomatic states, we constructed a regression model. The objective variable was the relative abundance of intestinal bacteria identified by the shotgun metagenomic analysis, and the dependent variables were the psychological, physical, and physiological index scores. As this was an exploratory study with a small sample size, we focused more on the effect size and did not conduct $p$-value correction in the analysis. We set the significance of the regression model at $p < 0.05$ or when the effect size of the model was above moderate ($R^2 \geq 0.13$).

## Reporting summary
Further information on research design is available in the Nature Portfolio Reporting Summary linked to this article.

## Data availability
All sequencing data has been deposited to the NCBI Sequence Read Archive under the project and are publicly available. The BioProject accession numbers are as follows: PRJNA844514 (16 s rRNA gene sequencing) for Study 1, PRJNA844516 (16 s rRNA gene sequencing and metagenomic shotgun sequencing) for Study 2. All questionnaire-related and physiological (i.e., physical composition, autonomic nervous system, and oxytocin hormone) are available from the Figshare database (https://doi.org/10.6084/m9.figshare.20439837).

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

## Acknowledgements

We are grateful to Daisuke Motooka (Genome Information Research Center, Research Institute for Microbial Diseases, Osaka University) for conducting the shotgun metagenomic analysis. This study was supported by a Grant-in-Aid for Scientific Research from Japan Society for the Promotion of Science (JSPS) (17H01016 to M. Myo. and 19K21813 to M. Myo.); a Grant-in-Aid for JSPS Fellows (19J15173 and 22J01448 to M. M.); a grant from the Center of Innovation Program, Japan Science and Technology Agency (JPMJCE1307 to M. Myo.); and the Japan Science and Technology Agency grants (JPMJMS2296 to M. Myo.).

## Author contributions

Conceptualization: M.M., M.T., K.H., M.Myo.; statistical analysis: M.M., M.T., S.W.; behavior assessment and analysis: M.M., M.T.; microbiome analysis: S.W., A.T.; hormone analysis: T.K., K.M., M.N.; writing−original draft: M.M.; writing−review & editing: M.M., M.T, S.W., A.T., T.K., K.M., M.N., K.H., M.Myo.; visualization: M.M., M.T; supervision: M.Myo., K.H.; funding acquisition: M.Myo., K.H., T.K.

## Competing interests

We declare we have no competing interests.
