## [Peer Review File · Communications Biology]

Reviewers' comments:

Reviewer #1 (Remarks to the Author):

1. The major finding of this study is that parenting stress in Japanese mothers is associated with their intestinal microbial dysbiosis, their autonomic nervous system, and their physical condition. The authors are claiming that mothers with high parenting stress have significantly reduced microbial diversity, abnormal short-chain fatty acid-producing bacteria, and increased inflammatory microbiota.
2. Overall, it is a well-written manuscript.
3. The study is novel as it was done first time on Japanese mothers. However, it is an observational study and only correlation is established. A longitudinal study with probiotic treatment would help to strengthen the relationship.
4. The study is cross-sectional and carried out for a short period of time. The fecal samples are collected only once. For studying intestinal microbiota, it is important to have at least two point samples (at the beginning and end). It is because the microbiota gets changed for several reasons apart from diet and stress.
5. The first study among the two is statistically well-planned.
6. There is a typo on line 521- "beat diversity"

Reviewer #2 (Remarks to the Author):

The study utilizes two cohort studies to investigate the relationship between maternal faecal microbiome and parenting stress. The first study utilizes a large cohort study "The Principle of Human Social Brain-Mind Development" with 339 participants. Study 2 utilizes another cohort study Investigation of the Interrelationship among Mental, Physical, and Intestinal Microbiota in Postpartum Mothers and Their Infants. Investigating 27 first time mothers. The authors found significant association between reported parental stress and changes in the alpha and beta diversity of the gut microbiome of the mothers. In particular, the presence of and butyrate-producing bacteria (e.g., *Lachnospira*).

Study 2 also investigated the association of the microbiome composition with autonomic nervous system function (oxytocin), physical condition (e.g., body composition, physical function, and physical condition), and psychological state (questionnaires).

The study methodology appears to be sound and of high scientific quality. The sample collection, microbiome analysis and statistical investigations are sound and of high scientific quality. Both questionnaires/surveys used to assess the psychological and physical well being of the mothers appear robust and well validated.

Results are appropriately displayed, please see comment below regarding table 1.

The discussion is in depth and considered. Care must be given to attributing causation rather than association to the presence of higher relative abundances of particular bacteria, however the assumptions are well supported by the references.

Conclusions appropriate.

This study is of high quality and adds to the body of literature in the field, it is novel in that it seeks to describe markers of resilience.

1. Line 20 suggest that "functioning" be changed to health
2. Line 30 change "cannot be restrained." to " is likely to continue to increase"
3. Table 1 needs to be re-formatted as it is very busy and difficult to read. Perhaps if the Shannon alpha diversity for each organism is displayed in Figure 2 then this information could be removed from table 1. Lists of organisms are perhaps better for an appendix and visual representation more

powerful for a general audience.

4. The structure of the discussion should be changed for clarity. Perhaps discussing each of the features as separate paragraphs would improve the readability. For instance rather than referring to feature 1 the paragraph (line 318) could begin, "abnormal levels of short chain fatty acid producing bacteria have been reported by others. "

5. Line 305 change "conditions" to "condition"

6. Line 312 change "less" to "lower"

7. Line 338 start sentence with "Study 2 was designed to we focus on primiparous mothers"

8. Line 344 the statement is confusing: "the diversity of the intestinal microbiota could contribute to psychological resilience through individual differences in vagal activity." Consider revising, as it is unclear if the vagal activity is the bacteria or the mother.

9. Line 348 "explained the individual differences" should be was associated with as there is no evidence for causation

10. Line 378 consider revising the introduction to "limitations of the study"

Reviewer #3 (Remarks to the Author):

The authors have presented some intriguing and novel findings connecting features of the gut microbiome with psychological and physical health among two distinct samples of Japanese mothers. Although these findings are just cross-sectional, I believe that they have the potential to lay the groundwork for future work in this area. As a side note, I was shocked by the finding that many of the new mothers in Study 2 had sarcopenia and poor physical function (e.g., grip strength), and this finding is certainly worthy of further exploration. There are several major and minor points that I hope the authors will address to increase the scientific soundness of this manuscript:

Major points:

(1) I do not see hypotheses specified anywhere. As a result, many of the results that are reported seem haphazard, as if the authors are just reporting whatever they happened to find that was significant. For example, in study 1, I don't see any findings reported for the correlations between the gut microbiome and physical functioning even though that appears to be a major focus of this article (and Study 2). In fact, physical functioning was not given much focus in the introduction and it appears to come out of the blue in the results section. Please rework the introduction and results section to show a coherent line of inquiry and specific hypotheses. The results section should then mirror these specific hypotheses.

(2) There are no covariates, which is problematic given that many of these associations (e.g., HRV and the microbiome) may be highly confounded by other factors like general health, exercise, etc.

(3) What is "nonclinical" in reference to the mothers? May want to use a more specific term in the abstract. How was it determined that they were free of psychiatric and physical disease?

(4) In study 1, how was the age range of 0-4 years old determined? Was this study specifically designed for these analyses, or are these secondary analyses?

(5) How was the specific vagal activity metric chosen? How was respiratory rate controlled for? Please provide more specifics about what participants were doing just prior to and during the heart rate recording. Were they seated, lying down, standing up? Were they holding an infant?

(6) Many important methodological details are only reported in the supplemental material, but they should be moved to the main text for reproducibility. Also critical details about the gut microbiome methods are not included in this manuscript (just a citation), but they should be included so that this could be a standalone manuscript.

(7) I do not see mothers' marital status reported anywhere, but this is an important factor in determining their parenting stress levels. In fact, it would be a good covariate.

(8) The group of physiological factors/physical functioning variables seem to be somewhat randomly chosen. Please provide more rationale and justification. Please provide citations about the predictive values of grip strength and gait speed. You may be interested in these recent articles concerning the link between HRV and the gut microbiome:

Mörkl, S., Oberascher, A., Tatschl, J. M., Lackner, S., Bastiaanssen, T. F., Butler, M. I., ... & Holasek, S. J. (2022). Cardiac vagal activity is associated with gut-microbiome patterns in women—An exploratory pilot study. *Dialogues in Clinical Neuroscience*, 24(1), 1-9.

Mörkl, S., Oberascher, A., Tatschl, J. M., Lackner, S., Bastiaanssen, T. F., Butler, M. I., ... & Holasek, S. J. (2023). The microbe-heart-brain dialogue: Vagal activity is associated with gut-microbiome patterns in women. *Journal of Affective Disorders Reports*, 12, 100553.

(9) Why were only mothers included and not fathers?

(10) It seems strange that only significant findings were only reported in text and null findings are reported in supplemental material. For example, in study 2, the null relationship between the gut microbiome and parenting stress should be reported in the main text given that it is a major focus of this manuscript. Also please comment on why this association might be null in Study 2 but not Study 1.

(11) Was the finding of a negative association between oxytocin and resilience hypothesized?

(12) Please provide a citation to support the relative within-person stability of the microbiome over the 2-3 day period that you have between the study visit and the stool sample collection.

(13) Please specify whether you tested relationships between all taxa in the microbiota or just the SCFA producers. Either way, there were many statistical tests run, but it appears that you only did a multiple test correction in Study 1, which could be problematic. Do any significant results survive in Study 2 with a multiple test correction?

Minor points:

(1) Please explain in the introduction why SCFA producers are important in terms of the gut-brain axis.

(2) Line 93 should read alpha diversity rather than just alpha.

(3) In line 110, please provide definitions of acronyms PSI-P and PSI-C.

(4) Have any other studies shown high risk of sarcopenia and poor physical functioning in new mothers?

(5) Line 247 should read "partially explained" rather than "explained"

(6) The authors mention certain facets of Japanese life (e.g., diet) that might be relevant to the variables of interest. Please comment in the limitations section about how these results may have been influenced by the fact that this was conducted in Japan.

(7) Figure 5 is very nice.

(8) Please define acronyms the first time they are presented in the main text.

(9) You may want to use this article to support the fact that you used V1V2 of 16s:

Kameoka, S., Motooka, D., Watanabe, S. et al. Benchmark of 16S rRNA gene amplicon sequencing using Japanese gut microbiome data from the V1-V2 and V3-V4 primer sets. *BMC Genomics* 22, 527 (2021). <https://doi.org/10.1186/s12864-021-07746-4>

Response to Reviewers

We are grateful to the reviewers for their thorough evaluation of our manuscript and providing valuable comments. We have modified the manuscript accordingly.

In the revised manuscript, the changes made are noted in red font color.

We hope you will find the revised manuscript suitable for publication in your esteemed journal.

Reviewer #1:

1. The major finding of this study is that parenting stress in Japanese mothers is associated with their intestinal microbial dysbiosis, their autonomic nervous system, and their physical condition. The authors are claiming that mothers with high parenting stress have significantly reduced microbial diversity, abnormal short-chain fatty acid-producing bacteria, and increased inflammatory microbiota.

2. Overall, it is a well-written manuscript.

3. The study is novel as it was done first time on Japanese mothers. However, it is an observational study and only correlation is established. A longitudinal study with probiotic treatment would help to strengthen the relationship.

Response to 1-3

Thank you for reviewing our manuscript. We recognize that a longitudinal study with probiotic treatment would be preferable to this correlational study. We hope to conduct a longitudinal research on this topic in the future.

=====

4. The study is cross-sectional and carried out for a short period of time. The fecal samples are collected only once. For studying intestinal microbiota, it is important to have at least two point samples (at the beginning and end). It is because the microbiota gets changed for several reasons apart from diet and stress.

4. Response

We acknowledge the importance of multiple sample points for studies of microbiota. However, to meet our specific research objectives, this was intentionally designed as a cross-sectional study that

observed and analyzed the microbiota at a single time point. Also, with over 300 participants, our study provides a robust dataset even with a single sample per individual. Additional sampling would have incurred substantial costs and logistical challenges, which we carefully weighed against the benefits during the design phase. We believe our methodology is justified given our research aims and resource considerations. For your reference, here are some other examples of cross-sectional gut microbiome studies with a single sample from each individual.

Osakunor, D.N.M., Munk, P., Mduluzi, T., et al. The gut microbiome but not the resistome is associated with urogenital schistosomiasis in preschool-aged children. *Commun Biol* **3**, 155 (2020). <https://doi.org/10.1038/s42003-020-0859-7>

Radjabzadeh, D., Bosch, J.A., Uitterlinden, A.G., et al. Gut microbiome-wide association study of depressive symptoms. *Nat Commun* **13**, 7128 (2022). <https://doi.org/10.1038/s41467-022-34502-3>

5. The first study among the two is statistically well-planned.

5. Response

We thank you for this positive and encouraging feedback.

6. There is a typo on line 521- “beat diversity”

6. Response

We thank the reviewer for pointing this out. We have corrected the typo.

Reviewer #2:

The study utilizes two cohort studies to investigate the relationship between maternal faecal microbiome and parenting stress. The first study utilizes a large cohort study “The Principle of Human Social Brain-Mind Development” with 339 participants. Study 2 utilizes another cohort study Investigation of the Interrelationship among Mental, Physical, and Intestinal Microbiota in Postpartum Mothers and Their Infants. Investigating 27 first time mothers. The authors found significant association between reported parental stress and changes in the alpha and beta diversity

of the gut microbiome of the mothers. In particular, the presence of and butyrate-producing bacteria (e.g., Lachnospira).

Study 2 also investigated the association of the microbiome composition with autonomic nervous system function (oxytocin), physical condition (e.g., body composition, physical function, and physical condition), and psychological state (questionnaires).

The study methodology appears to be sound and of high scientific quality. The sample collection, microbiome analysis and statistical investigations are sound and of high scientific quality. Both questionnaires/surveys used to assess the psychological and physical well being of the mothers appear robust and well validated.

Results are appropriately displayed, please see comment below regarding table 1.

The discussion is in depth and considered. Care must be given to attributing causation rather than association to the presence of higher relative abundances of particular bacteria, however the assumptions are well supported by the references.

Conclusions appropriate.

This study is of high quality and adds to the body of literature in the field, it is novel in that it seeks to describe markers of resilience.

1. Line 20 suggest that “functioning” be changed to health

1. Response

Thank you for this advice. We have changed “functioning” to “health.”

=====

2. Line 30 change “cannot be restrained.” to “ is likely to continue to increase.”

2. Response

Thank you for this advice. We have changed the content accordingly (*Lines 29–30*).

=====

3. Table 1 needs to be re-formatted as it is very busy and difficult to read. Perhaps if the Shannon alpha diversity for each organism is displayed in Figure 2 then this information could be removed from table 1. Lists of organisms are perhaps better for an appendix and visual representation more powerful for a general audience.

3. Response

Thank you for your feedback. The Shannon alpha diversity information was originally displayed in Figure 2. We agree that the information in Figure 2 and Table 1 overlap. Therefore, we have kept the results for the psychological and physical indices shown in Table 1 in the main text, moved the intestinal microbiota results to the Supplementary Materials, and modified the table to make it more comprehensible.

4. The structure of the discussion should be changed for clarity. Perhaps discussing each of the features as separate paragraphs would improve the readability. For instance, rather than referring to feature 1 the paragraph (line 318) could begin, “abnormal levels of short chain fatty acid-producing bacteria have been reported by others. “

4. Response

Thank you for helping us to clarify our line of thought. In the revised manuscript, we have modified the paragraph in question as you have suggested to improve readability as follows.

Lines 363–364: Regarding the first feature (i), abnormal levels of short-chain fatty acid-producing bacteria have been reported by previous a study.

Lines 377–378: Regarding the second and third features (ii and iii), previous research has found associations between psychiatric disorders and immunity and inflammation-related intestinal bacteria.

5. Line 305 change “conditions” to “condition.”

6. Line 312 change “less” to “lower.”

7. Line 338 start sentence with “Study 2 was designed to focus on primiparous mothers.”

5-7. Response

We thank the reviewer for this comment. We have made the advised alterations in *Lines 350, 357, and 385*, respectively.

8. Line 344 the statement is confusing: “the diversity of the intestinal microbiota could contribute to psychological resilience through individual differences in vagal activity.” Consider revising, as it is unclear if the vagal activity is the bacteria or the mother.

8. Response

Thank you for your feedback. We have modified this statement for improved clarity as follows.

Lines 393–395: “Assessing individual differences in both gut microbiota and vagal activity may help to clarify the physiological factors that contribute psychological resilience.”

9. Line 348 “explained the individual differences” should be was associated with as there is no evidence for causation

10. Line 378 consider revising the introduction to “limitations of the study”

9-10. Response

We agree with this suggestion and have altered the text in *Lines 395, 429*, respectively.

Reviewer #3:

The authors have presented some intriguing and novel findings connecting features of the gut microbiome with psychological and physical health among two distinct samples of Japanese mothers. Although these findings are just cross-sectional, I believe that they have the potential to lay the groundwork for future work in this area. As a side note, I was shocked by the finding that many of the new mothers in Study 2 had sarcopenia and poor physical function (e.g., grip strength), and this finding is certainly worthy of further exploration. There are several major and minor points that I hope the authors will address to increase the scientific soundness of this manuscript:

Major points

(1) I do not see hypotheses specified anywhere. As a result, many of the results that are reported seem haphazard, as if the authors are just reporting whatever they happened to find that was significant. For example, in study 1, I don't see any findings reported for the correlations between the gut microbiome and physical functioning even though that appears to be a major focus of this article (and Study 2). In fact, physical functioning was not given much focus in the introduction and it appears to come out of the blue in the results section. Please rework the introduction and results section to show a coherent line of inquiry and specific hypotheses. The results section should then mirror these specific hypotheses.

1. Response

In the original manuscript, we tried to keep the Introduction simple and concise. However, as this has provided insufficient information, we have made considerable revisions to this section. In particular, per the reviewer's suggestion, we have added to our descriptions of physical functioning, the autonomic nervous system, and relevant hormones. We have also revised the hypothesis to make it more explicit and have attempted to improve the clarity of the Introduction and Results. We have also cited the paper recommended in comment (8).

=====

(2) There are no covariates, which is problematic given that many of these associations (e.g., HRV and the microbiome) may be highly confounded by other factors like general health, exercise, etc.

2. Response

General health was controlled to some extent because only healthy mothers who were not currently hospitalized due to physical or mental illness were included. In addition, as the cohort consisted only of Japanese mothers of children aged 0–4 years, it is to be expected that our participants would share more sociodemographic characteristics with one another than would a more general adult sample (of mixed gender and diverse age groups). However, we agree that we should have attempted to examine covariates. Therefore, we reanalyzed maternal years of education and maternal age as covariates potentially related to health status and dietary habits, which were valid indexes for covariates among the measured variables. We found the results to be unchanged (Study 1). The results of this analysis are reported in the main text and the Supplementary Materials. For Study 2, we would have liked to incorporate covariates, but due to the small sample size, it is likely that the results would have been statistically invalid with their inclusion. We have explained this in the

limitations subsection and have recommended the examination of covariates (e.g., maternal body composition, dietary habits, exercise habits, etc.) in future studies.

Lines 466–468: “In future research with larger samples, it would be helpful to examine covariates, such as the body composition, dietary habits, and exercise habits of the mothers, to better understand the physical and mental effects of the intestinal microbiota.”

(3) What is "nonclinical" in reference to the mothers? May want to use a more specific term in the abstract. How was it determined that they were free of psychiatric and physical disease?

3. Response

Thank you for these questions. To avoid confusion, we have changed the term “nonclinical” in the Abstract to “healthy mothers with no physical or psychiatric disorders.” The presence of mental and physical diseases was determined using the Mykinso-Pro Questionnaire, which also gathers information on current and past treatments and medications. In the original manuscript, these data were provided in the Supplementary Materials because of the journal’s stipulated word limit. However, we have now moved it to the Methods section of the revised manuscript.

(4) In study 1, how was the age range of 0-4 years old determined? Was this study specifically designed for these analyses, or are these secondary analyses?

4. Response

We thank the reviewer for raising these questions. As we required the children in this study to be aged 0–4 years, we asked nursery schools and kindergartens to assist in the recruitment of research participants. This ensured that participants were mothers raising preschool children.

(5) How was the specific vagal activity metric chosen? How was respiratory rate controlled for? Please provide more specifics about what participants were doing just prior to and during the heart rate recording. Were they seated, lying down, standing up? Were they holding an infant?

5. Response

There are two major indices of heart rate variability that are believed to reflect vagal activity in the parasympathetic nervous system. The first of these is respiratory sinus arrhythmia (RSA) or high

frequency (HF) ratio based on frequency analysis; the second is cardiac vagal index (CVI) based on time-domain analysis. As noted, it is important to measure and control the respiratory rate when assessing RSA or HF ratio as these indices are associated with respiratory variability. However, frequency analysis indices cannot strictly evaluate sympathetic or parasympathetic function alone and are easily affected by noise, such as body movement. Moreover, the results of analyses based on short-time measurements are inherently unstable. Evaluation of vagal function using CVI employs the Lorenz plot method, and Toichi et al. (1997) (citation number is 68 in the main manuscript) have confirmed that vagal activity is reflected even when nerve blockade via medication is used. This method is less sensitive to respiratory rate, and the results of short-time analysis are relatively stable. For these reasons, we selected CVI as the metric used in this study. The reasons for this choice of index were originally given in the Supplementary Materials but are now briefly described in the main manuscript.

Respiratory rates were not measured. To clarify the setting of the resting state, we have added the following information to the Methods section.

Lines 665–672: “The R-R interval of the infants was calculated using the last minute of ECG data from the 3-min resting period. During these resting measurements, the mother sat on the floor and the infant lay on the floor. The mother was asked to relax and spend time with her infant as she usually would at home. Respiration was not measured. We analyzed the cardiac vagal and sympathetic indices using a Lorenz plot analysis method.¹² Lorenz plots are more robust than frequency analysis for analyzing ECG data over a short period or in the presence of noise due to body movement. They also have the advantage of allowing separate calculation of sympathetic and vagal activities and are independent of respiration and posture.”

(6) Many important methodological details are only reported in the supplemental material, but they should be moved to the main text for reproducibility. Also, critical details about the gut microbiome methods are not included in this manuscript (just a citation), but they should be included so that this could be a standalone manuscript.

6. Response

We had originally provided most of the methodology in the Supplementary Materials due to word limit constraints. However, we have now obtained approval from the Editor to extend the word limit

of our Methods section and have added detailed descriptions of our procedures. This includes the methods used to gather intestinal microflora data.

(7) I do not see mothers' marital status reported anywhere, but this is an important factor in determining their parenting stress levels. In fact, it would be a good covariate.

7. Response

We thank the reviewer for bringing this matter to our attention. Unfortunately, data were not gathered on the marital status of the mothers in our study. Although we could possibly have inferred this information from the SES in the questionnaire, it would have been imprecise data, so it was not incorporated as a covariate in the present study. As mentioned in our response to comment (2), we reanalyzed our results to include each mother's age and years of education (variables that could also be related to marital status) as covariates and found that the results remained the same. These results are presented in the Supplementary Materials.

(8) The group of physiological factors/physical functioning variables seem to be somewhat randomly chosen. Please provide more rationale and justification. Please provide citations about the predictive values of grip strength and gait speed. You may be interested in these recent articles concerning the link between HRV and the gut microbiome:

Mörkl, S., Oberascher, A., Tatschl, J. M., Lackner, S., Bastiaanssen, T. F., Butler, M. I., ... & Holasek, S. J. (2022). Cardiac vagal activity is associated with gut-microbiome patterns in women—An exploratory pilot study. *Dialogues in Clinical Neuroscience*, 24(1), 1-9.

Mörkl, S., Oberascher, A., Tatschl, J. M., Lackner, S., Bastiaanssen, T. F., Butler, M. I., ... & Holasek, S. J. (2023). The microbe-heart-brain dialogue: Vagal activity is associated with gut-microbiome patterns in women. *Journal of Affective Disorders Reports*, 12, 100553.

8. Response

Thank you for helping us to clarify our thoughts on this aspect of our study. We focused on vagal activity in heart rate variability because it is the main source of gut-to-brain neural information in the gut-brain axis. The oxytocin hormone was a focal physiological measure in this study because it is an important moderator of attachment, love, empathy, and maternal instincts during child-rearing. It

is also important in emotional processing and stress. Although a relationship has been demonstrated between physical and mental functional variables in adults, such basic data have not been collected from prenatal and postpartum mothers. This is an omission that must be addressed by future research on women's mental and physical health. We have added this background information to the Introduction. We have added the references you have recommended and associated citations to our revised manuscript.

We apologize for the missing references on grip strength and gait speed. These have now been added to the References section (no. 38, 40, 41).

(9) Why were only mothers included and not fathers?

9. Response

Although we are also very interested in examining the resilience in fathers, our study was limited by financial constraints. If we can obtain a larger grant in the future, we would definitely like to assess fathers also.

(10) It seems strange that only significant findings were only reported in text and null findings are reported in supplemental material. For example, in study 2, the null relationship between the gut microbiome and parenting stress should be reported in the main text given that it is a major focus of this manuscript. Also please comment on why this association might be null in Study 2 but not Study 1.

10. Response

We thank the reviewer for this comment. We have moved our null results to the main manuscript.

Regarding your comment “*Also, please comment on why this association might be null in Study 2 but not Study 1,*” we also reported an association between parenting stress (PSI infant) and the intestinal microbiota (*Eggerthella lenta*) in Study 2.

Regarding the difference between the results of Study 1 and Study 2, a possible explanation could be that the participants in Study 2 had lower baseline stress and depression scores than those in Study 1. However, due to COVID-19, we were unable to fully examine the results of the two studies and they remained a preliminary analysis. To address this, we have added the following to the limitations subsection.

Lines 454–468: “In Study 2, a regression analysis was performed to examine the relationship between intestinal microbiota and physical/physiological and psychological states. To prevent unnecessary statistical tests, we performed several correlation analyses to preprocess the regression analysis data. Owing to the small sample size, we did not perform p-value correction or mediation analysis, owing to the possibility of type 2 error and failure to perform appropriate p-value correction. Although we acknowledged the possibility of a type 1 error, all results had moderate to large effect sizes, and our single regression analysis confirmed significant associations. We also reported all non-significant results and provided scatter plots and details of 95% confidence intervals. Another issue that arose due to the small sample size was the inability to include covariates in Study 2. In addition, Study 2 participants had lower average parenting stress and depression scores than Study 1 participants. As both Study 1 and Study 2 were cross-sectional, we could not evaluate intrapersonal changes before and after childbirth. Therefore, longitudinal studies with larger samples, including samples from clinical groups with high stress levels, are required to clarify the neurophysiological and psychological relationships. In future research with larger samples, it would be helpful to examine covariates, such as the body composition, dietary habits, and exercise habits of the mothers, to better understand the physical and mental effects of the intestinal microbiota.”

(11) Was the finding of a negative association between oxytocin and resilience hypothesized?

11. Response

Thank you for your question. Many positive results on the effects of higher oxytocin levels have been reported, including increased mother-child interaction and lower stress. However, in recent years, studies have also begun to identify negative effects associated with increased oxytocin when it is accompanied by high parenting stress, including elevated aggression and reduced parenting behavior. Therefore, the function of oxytocin is still under debate. For the purposes of this study, we assumed that oxytocin may have both positive and negative associations. Rather than a bilateral hypothesis of positive/negative associations, our hypothesis regarding oxytocin and resilience was exploratory. We have also added more details on oxytocin to the Introduction (*Line 80-85, 113-115*).

(12) Please provide a citation to support the relative within-person stability of the microbiome over the 2-3 day period that you have between the study visit and the stool sample collection.

12. Response

David et al. (2016) have shown that the overall stability of the bacterial community is generally high, except during enteritis infection or overseas travel.

David, L.A., Materna, A.C., Friedman, J., Campos-Baptista, M.I., Blackburn, M.C., Perrotta, A., et al. Host lifestyle affects human microbiota on daily timescales. *Genome Biol* **15**(7), R89 (2014). doi: 10.1186/gb-2014-15-7-r89. Erratum in: *Genome Biol* **17**(1), 117 (2016). PMID: 25146375; PMCID: PMC4405912.

(13) Please specify whether you tested relationships between all taxa in the microbiota or just the SCFA producers. Either way, there were many statistical tests run, but it appears that you only did a multiple test correction in Study 1, which could be problematic. Do any significant results survive in Study 2 with a multiple test correction?

13. Response

As described in the passage of the Methods shown below, we did not focus solely on SCFAs-producing bacteria but examined the relationships between all bacteria that fell under the relative abundance (RA).

Lines 727–731 Method—Bioinformatics analysis—(4) Statistical analysis

“Additionally, we analyzed the composition of the prevalent microbiota as the set of genus-level bacterial groups shared by at least 50% of the participants, with a relative abundance (RA) of at least 0.01%, based on previous studies.⁵⁵ We calculated RA as the number of sequencing reads of each taxon in a sample standardized by the total number of sequences generated for that sample.”

In Study 2, our objective was to perform a regression analysis to determine the extent to which individual differences in the intestinal microbiota could explain differences in physical and psychological resilience. It is common in psychology research to preprocess data with correlation analyses to minimize the number of tests before conducting regression analysis. As regression analyses are modeled one by one, it is not always absolutely necessary to correct the p -value. As mentioned in the limitations subsection of the Discussion, Study 2 had a very small sample size. In such cases, both the p -value and the effect size should be considered. Therefore, as a pre-processing measure before the regression analysis, we performed an exploratory correlation analysis focused on the variables with either $p < 0.05$ or a moderate effect size of $r \geq 0.30$.

However, we recognize that some may argue that more rigorous p -value correction should have been performed at this stage. After considering this, we adopted the above approach in this case. The main reason is that if a p -value correction like Bonferroni's is performed at the correlation analysis stage, the criteria are too strict and the possibility of type 2 error is much higher. However, in cases, such as Study 1, where there is correction by q -values, correction is based on the distribution of p -values. In Study 2, however, the number of indicators (18 items) for which correlation analysis was conducted for one bacterium was insufficient to produce a p -value distribution. This can produce incorrectly calculated q -values in some cases. As stated in our paper, Study 2 is reported as a preliminary study, and we mention in our discussion of the study limitations that our results require future replication with a larger sample.

In response to your point, we felt that the limitation in the main text could be misinterpreted by the reader because it was worded to describe multiple comparisons as succinctly as possible. We have modified this to provide a slightly more detailed description as follows.

Lines 454–461: “In Study 2, a regression analysis was performed to examine the relationship between intestinal microbiota and physical/ physiological and psychological states. To prevent unnecessary statistical tests, we performed several correlation analyses to preprocess the regression analysis data. Due to the small sample size, we did not perform p -value corrections or mediation analyses. Although we acknowledge the possibility of a type 1 error, all results had moderate to large effect sizes, and our single regression analysis confirmed significant associations. We also reported all non-significant results, and provided scatter plots and details of 95% confidence intervals.”

Minor points

(1) Please explain in the introduction why SCFA producers are important in terms of the gut-brain axis.

1. Response

In accordance with this suggestion, the following has been added to the Introduction.

Lines 38–43: “For example, a recent systematic review found that patients with depression and anxiety disorders have increased inflammatory intestinal bacteria and fewer short-chain fatty acid

(SCFA)-producing intestinal bacteria.¹⁰ SCFAs are produced by gut microbiota-induced fermentation of dietary fiber. Among the SCFAs, butyrate-producing bacteria in particular have many beneficial effects on the intestinal environment, such as improving the immune function of the intestinal mucosa and inhibiting cancer cells.”

- (1) (2) Line 93 should read alpha diversity rather than just alpha.
(3) In line 110, please provide definitions of acronyms PSI-P and PSI-C.

2-3. Response

Thank you for these suggestions. We have made the suggested modifications in the revised manuscript. Lines 131 and 149, respectively.

- (4) Have any other studies shown high risk of sarcopenia and poor physical functioning in new mothers?

4. Response

To the best of our knowledge, no studies to date have examined the risk of sarcopenia and loss of physical function in young women (or mothers specifically), because sarcopenia research has predominantly focused on older adults. Hence, the results of this study are important as they indicate a tendency to low muscle mass among postpartum women. This suggests that the risk of sarcopenia may begin at a younger age than previously assumed if muscle strength is not restored following pregnancy and childbirth. As mentioned in our response to comment (8), there is a need for further research concerned with the quantitative assessment of women's physical functions.

- (5) Line 247 should read "partially explained" rather than "explained"

5. Response

We agree with the reviewer and have modified this line accordingly. Line 289

(6) The authors mention certain facets of Japanese life (e.g., diet) that might be relevant to the variables of interest. Please comment in the limitations section about how these results may have been influenced by the fact that this was conducted in Japan.

6. Response

Thank you for this advice. As you have pointed out, intestinal microflora and dietary habits in Japan are unique and are not equivalent to those observed in other Asian countries. Food culture is an important contributor to the intestinal microbiota. Hence, it is important to establish whether the results of the present study are specific to Japan or common worldwide. In either case, future research is required to address the mechanisms responsible for the results demonstrated in this study.

We have added the following.

Lines 449–451: “Whether the results of this study are unique to data obtained in Japan or common worldwide is an important question that should be examined in future studies, both by accumulating further research on the Japanese, and by conducting international comparisons.”

=====
(7) Figure 5 is very nice.

7. Response

Thank you very much. We greatly appreciate your positive feedback.

=====
(8) Please define acronyms the first time they are presented in the main text.

8. Response

Thank you for reminding us of this requirement. We have checked the text and ensured all acronyms have now been defined upon first use.

=====
(9) You may want to use this article to support the fact that you used V1V2 of 16s:

Kameoka, S., Motooka, D., Watanabe, S. et al. Benchmark of 16S rRNA gene amplicon sequencing using Japanese gut microbiome data from the V1–V2 and V3–V4 primer sets. BMC Genomics 22, 527 (2021). <https://doi.org/10.1186/s12864-021-07746-4>

9. Response

Thank you for this suggestion. We have added the suggested reference to the References section and cited it in the relevant subsection of the Methods (*Line 719*).

REVIEWERS' COMMENTS:

Reviewer #1 (Remarks to the Author):

All the major concerns were addressed and explained in this revised version.

Reviewer #2 (Remarks to the Author):

I am satisfied with the authors responses to my comments

I believe the authors have responded adequately to reviewer 3.

Only comment would be change response:

Lines 466–468: "In future research with larger samples, it would be helpful to examine covariates, such as the body composition, dietary habits, and exercise habits of the mothers, to better understand the physical and mental effects of the intestinal microbiota."

To

Lines 466–468: "In future research with a larger sample sizes, it would be helpful to examine covariates, such as the body composition, dietary habits, and exercise habits of the mothers, to better understand the physical and mental effects of the intestinal microbiota."

Response to Reviewers

We are grateful to the reviewers for their thorough evaluation of our manuscript and providing valuable comments. We have modified the manuscript accordingly.

Reviewer #2:

Only comment would be change response:

Lines 466–468: “In future research with larger samples, it would be helpful to examine covariates, such as the body composition, dietary habits, and exercise habits of the mothers, to better understand the physical and mental effects of the intestinal microbiota.”

To

Lines 466–468: “In future research with a larger sample sizes, it would be helpful to examine covariates, such as the body composition, dietary habits, and exercise habits of the mothers, to better understand the physical and mental effects of the intestinal microbiota.”

Response

Thank you for reviewing our manuscript. We have changed our manuscript as the reviewer 2 suggested.